# Fairness via Representation Neutralization

**Mengnan Du[1]*, Subhabrata Mukherjee[2], Guanchu Wang[3], Ruixiang Tang[3],**
**Ahmed Hassan Awadallah[2], Xia Hu[3]**
[1]Texas A&M University    [2]Microsoft Research    [3]Rice University
dumengnan@tamu.edu, {submukhe,hassanam}@microsoft.com
{guanchu.wang,rt39,xia.hu}@rice.edu

## Abstract

Existing bias mitigation methods for DNN models primarily work on learning debiased encoders. This process not only requires a lot of instance-level annotations for sensitive attributes, it also does not guarantee that all fairness sensitive information has been removed from the encoder. To address these limitations, we explore the following research question: *Can we reduce the discrimination of DNN models by only debiasing the classification head, even with biased representations as inputs?* To this end, we propose a new mitigation technique, namely, Representation Neutralization for Fairness (RNF) that achieves fairness by debiasing only the task-specific classification head of DNN models. To this end, we leverage samples with the same ground-truth label but different sensitive attributes, and use their neutralized representations to train the classification head of the DNN model. The key idea of RNF is to discourage the classification head from capturing undesirable correlation between fairness sensitive information in encoder representations with specific class labels. To address low-resource settings with no access to sensitive attribute annotations, we leverage a bias-amplified model to generate proxy annotations for sensitive attributes. Experimental results over several benchmark datasets demonstrate our RNF framework to effectively reduce discrimination of DNN models with minimal degradation in task-specific performance.

## 1 Introduction

Deep neural networks (DNNs) have made significant advances in recent times [1, 2, 3], and have been deployed in many real-world applications. However, DNNs often suffer from biases and show discrimination towards certain demographics, especially in high-stake applications, such as criminal justice, employment, loan approval, credit scoring, etc [4, 5, 6]. For example, COMPAS, an algorithmic recidivism predictor, is likely to associate African-American offenders with higher risk scores compared to Caucasians while having a similar profile [7]. This brings significant harm to both society and individuals, thus leading to recent focus on mitigation techniques to alleviate the adverse effects of DNN biases.

Existing debiasing methods usually work on *learning debiased representations* at the encoder-level. One representative family of methods perform mitigation by explicitly learning debiased representations, either through adversarial learning [8, 9, 10] or invariant risk minimization [11, 12]. Another family of methods [13, 14, 15] implicitly learn debiased representations by incorporating explanation during model training to suppress it from paying high attention to biased features in the original input. Essentially, the above methods aim to remove the bias from deep representations.

Learning debiased representations is a technically challenging problem. Firstly, it is hard to remove all fairness sensitive information in the encoder. The suppression of fairness sensitive information

---

* Part of the work was done while the first author was an intern at Microsoft Research.

might also remove useful information that is task relevant. Secondly, most existing debiasing methods assume access to additional meta-data such as fairness sensitive attributes and a lot of annotations corresponding to the protected groups to guide the learning of debiased representations. However, such resources are expensive to obtain, if not unavailable, for most real world applications.

To address these limitations, we explore the following research question: *Can we reduce the discrimination of DNN models by only debiasing the task-specific classification head, even with a biased representation encoder?* Our work is motivated by the empirical observation that standard training can result in the classification head capturing undesirable correlation between fairness sensitive information and specific class labels. Some recent works [16, 17, 18] have explored such spurious or shortcut learning behavior of DNNs in various applications. To this end, we propose the RNF (Representation Neutralization for Fairness) framework for mitigation, motivated by the Mixup work [19, 20]. We first train a biased teacher network via standard cross entropy loss. In the second stage, we freeze the representation encoder of the biased teacher, and only update the classification head via representation neutralization. This discourages the model from associating biased features with specific class labels, and enforces the model to focus more on task relevant information. To address low-resource settings, our RNF framework does not require any access to the protected attributes during training. To this end, we train a bias-amplified model using generalized cross entropy loss that is used to generate proxy annotations for sensitive attributes. Experimental results over several benchmark tabular and image datasets demonstrate our RNF framework to significantly reduce discrimination of DNN models with minimal degradation of the task performance. The major contributions of our work can be summarized as follows:

- We analyze bias propagation from the encoder representations to the final task-specific layer demonstrating that DNN models heavily rely on undesirable correlations for prediction.
- We introduce RNF, a bias mitigation framework for DNN models via representation neutralization. Our RNF framework achieves mitigation without any access to instance-level sensitive attribute annotations, and instead relies on self-generated proxy annotations.
- Experimental results on several benchmark datasets demonstrate the effectiveness of our RNF framework via debiasing only the classification head while using biased representations as input. Additionally, we show RNF to be complementary to existing methods that learn debiased encoders and can be further improved within our framework.

## 2 Related Work

We briefly review bias mitigation and broader robustness literature which are most relevant to ours.

**Bias Mitigation.** Recent studies have indicated that DNN models exhibit social bias towards certain demographic groups. This has led to increased attention to bias mitigation techniques in recent times [21, 22, 23]. Existing mitigation methods can be generally grouped into three broad categories. The first one is based on adversarial training [8, 9, 10]. It leads to a fair classifier as the predictions cannot carry any group discrimination information that the adversary can exploit. However, this method assumes that the sensitive attribute annotations are known, and uses those annotations to learn debiased representations. The second representative family of mitigation methods is based on explainability [13, 14, 15, 24]. These methods require fine-grained feature-level annotations to specify which subset of features are fairness sensitive. The third category falls under the umbrella of causal fairness [25, 11, 12]. The main idea is to enforce the model to concentrate more on task relevant causal features, and getting rid of superficial correlations [26]. This results in the model capturing debiased representations. For instance, [27] minimizes the correlation between sentence representations and bias words using a contrastive learning framework. We can typically decouple the classification problem into representation learning and classification as the two major parts [28]. Different from the above-mentioned methods that mainly aim to train the debiased representations, our work focuses on the *classification head* and aims to suppress it from capturing undesirable correlation between fairness sensitive attributes and class labels.

**Shortcut Learning in DNNs.** Our work is motivated by the observation that DNNs with standard training are prone to exploit undesirable correlations (or shortcuts in the dataset) for prediction [16, 29]. Beyond algorithmic discrimination, recent studies show that *shortcut learning* [17] can result in other undesirable consequences like poor generalization and adversarial vulnerability. Specifically, this leads to a high performance degradation for previously unseen inputs, especially for those data beyond

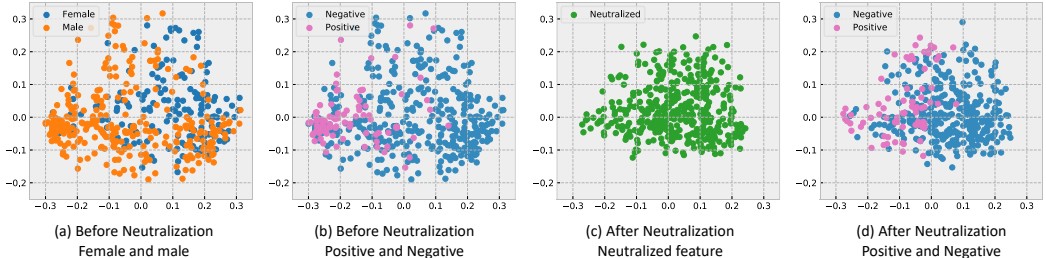

| (a) Before Neutralization
Female and male | (b) Before Neutralization
Positive and Negative | (c) After Neutralization
Neutralized feature | (d) After Neutralization
Positive and Negative |

Figure 1: Representation analysis of $z$ using Kernel PCA. (a) Protected attribute $a$ (i.e., gender) is a discriminative feature. In this plot, the male group primarily lies in the lower left interval, whereas the female group is located primarily on the upper right interval. (b) Predicted positive label and negative task-label distribution. (c) Representation neutralization (please refer to Sec. 3.3) to reduce the discriminative power of $a$ for dimensions relevant to the sensitive information, comparing to the distribution in (a). (d) Neutralization still preserves the useful task relevant information.

held-out test set. Representative tasks depicting such behavior include reading comprehension [30], natural language inference [18], visual question answering[31] and deepfake detection [32].

The most similar work to ours also regularizes the model training using interpolated samples between groups [33]. Similar to the aforementioned solutions, this work also heavily relies on sensitive attribute annotations in the training set. This limits the application scenarios of the mitigation algorithms, especially for many real-world applications without readily available annotations dealing with sensitive attributes. Our representation neutralization framework achieves comparable or better performance to such techniques without relying on such sensitive annotations by leveraging proxy annotations obtained from a bias-intensified version of our framework, thereby, making it broadly applicable to arbitrary real-world applications.

## 3 Representation Neutralization for Fairness

In this section, we first analyze the task-specific classification head of a DNN to examine how bias is propagated from the encoder representation layer to the task-specific output layer (Section 3.2). We empirically demonstrate the undesirable correlation between fairness sensitive information in representations with specific class labels. Based on the observation, we introduce the Representation Neutralization for Fairness (RNF) framework to debias DNN models (Section 3.3). Finally, we propose an approach to generate proxy annotations for sensitive attributes, enabling the RNF framework to be applicable to low-resource settings with no access to sensitive attribute annotations (Section 3.4).

### 3.1 Notations

We first introduce the notations used in this work. Let $\mathcal{X} = \{x_i, y_i, a_i\}, i \in 1, ..., N$ be the training set, where $x_i$ is the input feature, $y_i$ denotes the ground truth label, and $a_i$ represents the sensitive attribute (e.g., gender, race, age). For ease of notation, in the following sections, we consider binary sensitive attributes[2]. Nevertheless, *our proposed mitigation framework can also be applied to non-binary sensitive attributes (e.g., age)*.

Consider the classification model $f(x, \theta) = c(g(x))$, parameterized with $\theta$ as the model parameters. Here $g(x) : \mathcal{X} \to \mathcal{Z}$ represents the feature encoder, and $g(x) = z$ is the representation for $x$ obtained from a DNN model. The predictor $c(z) : \mathcal{Z} \to \mathcal{Y}$ is the *multi-layer* classification head. It is depicted by the top layer(s) of the DNN, which takes the encoded representation $z$ as input and maps it to softmax probability. The final class prediction is denoted by $\tilde{y} = \arg\max c(z)$. In this work, we aim to reduce the discrimination of DNN models by *only debiasing the classification head $c(z)$*, with the biased representation encoder $g(x)$ as input.

---

[2]Gender is not binary in reality as there are many different gender identities, such as male, female, transgender, to name a few. In this work, we consider gender as a binary sensitive attribute due to the limitation of the benchmark dataset that encodes gender as a binary variable.

## 3.2 Analysis of the Classification Head

In this section, we examine how bias manifests in the representation space $\mathcal{Z}$ as well as how the classification head $c(z)$ obtained with standard training scheme propagates bias from the representation layer to the model output layer. To this end, we train a biased network $f_T(x)$ via standard cross entropy loss, where the following experiment is performed using the Adult dataset [34].

**Representation Probing Analysis.** For the Adult training set, we generate representation vectors for 500 training samples using the biased network $f_T(x)$ and project them in 2D for visualization. To this end, we utilize the Kernel Principal Component Analysis (KPCA) [35] with a sigmoid kernel, which is a tool to visualize high-dimensional data. Since the classification head $c(z)$ contains multiple non-linear layers, we choose kernel PCA instead of a linear dimensionality reduction method such as linear PCA. The visualization is shown in Figure 1 (a). The plot depicts that the low dimensional projection separates the two protected groups in two areas, where the male group is primarily located in the lower left area, whereas the female group primarily occupies the upper right area. Comparing the task-label $y$ distribution in Figure 1 (b) with the protected group distribution in Figure 1 (a), we observe that the protected attribute information is a discriminative feature that could be exploited by the task classification head for prediction.

**Role of the Biased Classification Head.** The above demonstrative analysis indicates that the model representation captures both useful task relevant classification information as well as bias information from protected attributes. Specifically, the model captures strong correlation between the fairness sensitive information and the class labels. On analyzing the data distribution, we observe this to be an artifact of the conditional label distribution with respect to the sensitive attributes being skewed. The model relies on this shortcut for prediction, resulting in bias amplification. We observe the male neurons to positively correlate to the desirable label (also refer to the experimental analysis in Sec. 4.3), whereas the female neurons positively correlate to undesirable label. This depicts an *undesirable correlation* between sensitive information with certain class labels in the model.

**Our Motivation.** Based on the above empirical observations, we propose to neutralize the training samples (Fig 1 (c)) so as to reduce the discriminative power of the fairness sensitive information, while at the same time preserving task relevant information (Fig 1 (d)). With the neutralized training data, we propose to re-train the classification head. Our goal is to adjust the decision boundary to implicitly de-correlate the fairness sensitive information in the representation space with class labels.

## 3.3 Representation Neutralization for Debiasing Classification Head

Based on the aforementioned empirical observations, in this section we propose a simple yet effective bias mitigation framework via Representation Neutralization for Fairness (RNF). RNF does not require any prior knowledge about existing bias in the representation space; nor does it require any knowledge about specific dimension(s) encoding the sensitive attributes – making it widely useful for arbitrary applications. Our goal is to encourage the model to ignore the sensitive attributes and instead focus more on task relevant information.

RNF is implemented in two steps (see Figure 2). In the first step, we train the model using cross entropy loss, and obtain a biased teacher network $f_T(x)$. During the second step, we freeze the encoder $g(x)$ for $f_T(x)$, and use it as our backbone encoder for learning representations. We then re-train only the classification head $c(z)$ using feature neutralization (see Figure 2 (b)).

**Representation Neutralization.** To this end, while training the classification head, for an input sample $\{x_1, y, a_1\}$, we *randomly select another sample* $\{x_2, y, a_2\}$, with the same class label $y$ but a different sensitive attribute $a_2$ compared to $a_1$ in the input sample. Now we compute the corresponding representations $z_1 = g(x_1)$ and $z_2 = g(x_2)$ and re-train the classification head using the neutralized representation $z = \frac{z_1 + z_2}{2}$ as input. For the supervision label $y$ for the classification head, we utilize the neutralized soft probability $y = \frac{p_1 + p_2}{2}$ after temperature scaling obtained as follows. Given the logit vector $z_1$ for input $x_1$, the probability for class $i$ is computed as $p_{1,i} = \frac{\exp(z_{1,i}/T)}{\sum_j \exp(z_{1,j}/T)}$, where $T \geq 1$. This can be regarded as a form of knowledge distillation [36], where $T > 1$ softens the softmax score. A larger temperature prevents the model from assigning over-confident prediction probabilities. A special case for RNF is when $T = 1$, where $p_1$ and $p_2$ are the standard softmax probabilities obtained from the biased teacher network $f_T(x)$.

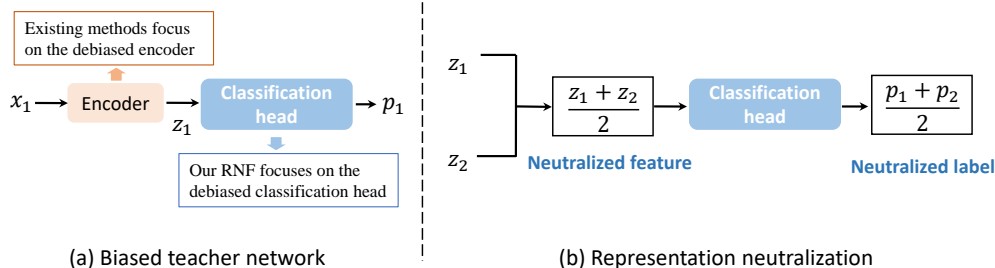

(a) Biased teacher network | (b) Representation neutralization

Figure 2: Debiasing with representation neutralization. (a) We first train a biased teacher network using only cross entropy loss. For two inputs $x_1$ and $x_2$ that with the same class label $y$ and different sensitive attribute $a$, we obtain the representations $z_1$ and $z_2$, and softened probabilities $p_1$ and $p_2$. (b) We freeze parameters of the biased encoder and only re-train the classification head using the neutralized representation $\frac{z_1+z_2}{2}$ as input, and softened probability $\frac{p_1+p_2}{2}$ as supervision signal.

We use the knowledge distillation loss. In particular, the mean squared error (MSE) loss is used as a distance-based metric to measure the similarity between model prediction and the supervision signal.

$$\mathcal{L}_{\text{MSE}} = (\hat{y}_i - y)^2 = \{c(\frac{1}{2}z_1 + \frac{1}{2}z_2) - (\frac{1}{2}p_1 + \frac{1}{2}p_2)\}^2. \tag{1}$$

where, $c$ is the classification head to project representations to softmax prediction probability.

There are two main benefits of the aforementioned training scheme. From the input perspective, the neutralization of representations suppresses the model from capturing the undesirable correlation between fairness sensitive information in the representation with the class labels. From the output perspective, the softened label encourages the model to assign similar predictions to different sensitive groups. Optimizing Eq. (1) can lead to reduced generalization gap between the two groups.

**Theorem 1** *Given a well-trained representation encoder $g(x) = z$ satisfying $||z_1 - z_2||_2 \leq \lambda_z |p_1 - p_2|$ for $x_1, x_2 \in \mathcal{X}$ and a bounded loss function $L(c(z_i), p_i) = (1 - p_i)l(c(z_i), y = 0) + p_i l(c(z_i), y = 1)$, where $l(c(z_i), y = j) \leq \epsilon_L$ for $x_i \in \mathcal{X}$ and $j = 0, 1$; if the classification head $c$ minimizes the loss between neutralized representation and soft probabilities, i.e. $\left\| \nabla_z L(c(z), p) \big|_{z=\frac{z_1+z_2}{2}, p=\frac{p_1+p_2}{2}} \right\|_2 \leq \epsilon_c$, where $z = g(x)$, $x_1 \sim P(x_1 \mid a_i = 0)$, $x_2 \sim P(x_2 \mid a_i = 1)$, $|p_1 - p_2| \leq \epsilon_p$, the gap of generalization loss between groups $a = 0$ and $a = 1$ is bounded by:*

$$\left| \mathbb{E}_{x_i \sim P(x_i|a_i=0)} L(c(z_i), p_i) - \mathbb{E}_{x_j \sim P(x_j|a_j=1)} L(c(z_j), p_j) \right| \leq \epsilon_p(\lambda_z \epsilon_c + \epsilon_L) \tag{2}$$

Here "well-trained" means that the model has learned reasonably good representations to encode sufficient task relevant information. Essentially, after training for a sufficient number of epochs until the validation loss has converged, we have access to a reasonably good representation space (although it might encode a lot of fairness sensitive information). If we use the RNF loss function to further train the classification head, the generalization gap between the different protected groups would be small. For more detailed proof, please refer to Section A in the Appendix.

**Smoothing Neutralization.** To further enforce the model to ignore sensitive attributes, we construct augmented training samples using a hyper-parameter $\lambda$ to control the degree of neutralization of the samples $\{z_1, p_1, y\}$ and $\{z_2, p_2, y\}$. The augmented neutralized sample is given by $z = \lambda z_1 + (1 - \lambda)z_2$, $\lambda \in [\frac{1}{2}, 1)$. We encourage the classification head to give similar prediction scores for the augmented and the neutralized sample (with $\lambda = \frac{1}{2}$). The regularization loss is given by:

$$\mathcal{L}_{\text{Smooth}} = \sum_{\lambda \in [\frac{1}{2}, 1)} |c(\lambda z_1 + (1 - \lambda)z_2) - c(\frac{1}{2}z_1 + \frac{1}{2}z_2)|_1. \tag{3}$$

By varying $\lambda$ we control the degree of sensitive information for the augmented samples. It is utilized to penalize the large changes in softmax probability when we move along the interpolation between two samples. We linearly combine the MSE loss in Eq. (1) with the regularization term as follows:

$$\mathcal{L} = \mathcal{L}_{\text{MSE}} + \alpha \mathcal{L}_{\text{Smooth}}. \tag{4}$$

We train the classification head using the loss function in Eq. (4). Eventually we combine the original encoder of $f_T(x)$ and re-trained classification head as the debiased student network $f_S(x)$. The teacher $f_T(x)$ is later discarded and the debiased student network $f_S(x)$ is used for prediction.

## 3.4 Generating Proxy Annotations for Sensitive Attributes

The aforementioned feature neutralization is limited in that it requires instance-level sensitive attribute annotations $\{a_i\}_{i=1}^N$ for all training samples. Such resource-extensive annotations are difficult to obtain for many practical applications particularly due to the nature of the sensitive attributes. To address this limitation, we propose a method to generate proxy annotations $\{\hat{a}_i\}_{i=1}^N$ for the sensitive attributes based on the model uncertainty. The key idea is that a biased model generates over-confident predictions for one demographic group, while giving much lower scores for the alternative group. Particularly, the bias-amplified model tends to assign the privileged group more desired outcome, while assigning the under-privileged group less-desired outcome. For instance in the Adult dataset, the average prediction probability of the desired label (higher income) for the male group is much higher than that of the female group. In contrast, the average probability of the less-desired label for the female group is much higher than that of the male group.

To better facilitate the model to generate uncertainty scores, we train another biased model by intentionally amplifying the bias via generalized cross entropy loss (GCE) [37]. The bias-amplified model is denoted as $f_B(x)$ and the loss function is given as follows:

$$\text{GCE}(f(x;\theta), y) = \frac{1 - f_y(x;\theta)^q}{q}, \tag{5}$$

where $f_y(x;\theta)$ denotes the output probability for ground truth label $y$. The hyper-parameter $q \in (0, 1]$ controls the degree of bias amplification. When $\lim_{q \to 0}$, the GCE loss in Equation (2) approaches $-\log p$ which is equivalent to standard cross entropy loss. The core idea is that for more biased samples, i.e., samples with larger $f_y(x;\theta)$ value, the model assigns higher weights $f_y^q$ while updating the gradient.

$$\frac{\partial \text{GCE}(p, y)}{\partial \theta} = f_y^q \frac{\partial \text{CE}(p, y)}{\partial \theta}. \tag{6}$$

In this setup, the model $f_B(x)$ learns more from bias-amplified samples compared to the model $f_T(x)$ trained with standard cross entropy loss.

The confidence score of $f_B(x)$ is used to indicate whether a sample belongs to a privileged or unprivileged group. Specifically, for a desired ground truth label, samples with over-confident scores are grouped into the privileged group, whereas subsets of samples with low prediction scores are grouped into the unprivileged group. In contrast, for the undesired ground truth label, samples with over-confident scores are grouped into the unprivileged group, and vice versa. Based on this criterion, we generate proxy sensitive attribute annotation $\hat{a}$ for each training sample $x$ to split samples into two groups, which are subsequently used for feature neutralization.

The overall process of our RNF mitigation framework is given in Algorithm 1, which contains two stages. In the first stage, we train the biased teacher network $f_T(x)$ and the bias-amplified network $f_B(x)$. In the second stage, we first use $f_B(x)$ to generate proxy sensitive attribute annotations for all training samples. We use the ratio $\gamma$ to partition the training set to generate proxy annotations for protected attributes that are subsequently used for feature neutralization. Note that the ratio $\gamma$ is determined by the fairness-accuracy trade-off on the validation set. Then we use representation neutralization and the loss function in Eq. (4) to re-train the classification head of $f_T(x)$. Eventually, we combine the original encoder $g(x)$ of $f_T(x)$ and the refined classification head $c(z)$ to give us the debiased student network $f_S(x)$. It is worth noting that the neutralization is merely performed during the training stage for only debiasing the classification

---

**Algorithm 1:** RNF mitigation framework.

**Input:** Training data $D = \{(x_i, y_i)\}_{i=1}^N$.
1 Set hyperparameter $q, \alpha$.
2 **while** *first stage* **do**
3     Train teacher network $f_T(x)$ and bias-amplified network $f_B(x)$.
4 **while** *second stage* **do**
5     Determine the splitting threshold $\gamma$ for $f_B(x)$;
    Calculate proxy annotation $\hat{a}_i$ for each training
6     sample $\{(x_i)\}_{i=1}^N$ based on $\gamma$ and $f_B(x)$;
    Use loss function in Eq. (4) to re-train
7     classification head $c(z)$ of $f_T(x)$.
**Output:** The debiased student network $f_S(x)$.

head. At inference time, the features just come in as encoded, and the classification head has learned to not exploit any of the information correlated with the sensitive attribute for prediction.

# 4 Experiments

In this section, we conduct experiments to evaluate the effectiveness of our RNF framework.

## 4.1 Experimental Setup

### 4.1.1 Fairness Metrics, Benchmark Datasets and Baselines

We use two group fairness metrics: Demographic Parity [38] and Equalized Odds [39]. Demographic Parity (DP) measures the ratio of the probability of favorable outcomes between unprivileged and privileged groups: $\text{DP} = \frac{p(\hat{y}=1|a=0)}{p(\hat{y}=1|a=1)}$. Equalized Odds ($\Delta$EO) require favorable outcomes to be independent of the protected class attribute $a$, conditioned on the ground truth label $y$. Specifically, it calculates the summation of the True Positive Rate difference and False Positive Rate difference:

$$\Delta\text{EO} = \{P(\hat{y}=1 \mid a=0, y=1) - P(\hat{y}=1 \mid a=1, y=1)\} + \{P(\hat{y}=1 \mid a=0, y=0) - P(\hat{y}=1 \mid a=1, y=0)\}. \quad (7)$$

Under the above metrics, it is desirable to have a DP value closer to 1 and $\Delta$EO value closer to 0.

We use two benchmark tabular datasets and one image dataset to evaluate the effectiveness of RNF. For the Adult income dataset (**Adult**), the goal is to predict whether a person's income exceeds \$50K/yr [34]. We consider *gender* as the protected attribute where vanilla trained models show discrimination towards the female group by predicting females to earn less. For the Medical

Table 1: Dataset statistics.

|  | Adult | MEPS | CelebA |
|---|---|---|---|
| # Training | 33120 | 11362 | 194599 |
| # Validation | 3000 | 1200 | 4000 |
| # Test | 9102 | 3168 | 8000 |

Expenditure dataset (**MEPS**), we consider two groups *white* and *non-white* [40]. Here the task is to predict whether a person would have a 'high' utilization, where vanilla DNN shows discrimination towards the non-white group. The CelebFaces Attributes (**CelebA**) dataset is used to predict whether the hair in an image is wavy or not [41]. We consider two groups *male* and *female*, where vanilla trained models show discrimination towards the male group. We split all datasets into three subsets with statistics reported in Table 1. More details of the datasets are included in Sec. C in the Appendix.

We compare two variants of our framework **RNF** (using *proxy* sensitive attribute annotations) and **RNF_GT** (using *ground truth* sensitive attribute annotations) against baselines such as DNNs trained using only cross entropy loss (referred as **Vanilla**) and two regularization based mitigation methods, namely, adversarial training (**Adversarial**) [42] and Equalized Odds Regularization (**EOR**) [43]. Among them, the Adversarial method achieves fairness via learning debiased representations, whereas EOR directly optimizes the EO metric in Eq. (7). All three baselines control the fairness-accuracy trade-off via hyper-parameters. More details on the baselines are included in Sec. D in the Appendix.

### 4.1.2 Implementation Details

For the image classification task, we use ResNet-18 [1] (we add one more fully connected layer). We set the representation encoder $g(x)$ as the convolutional layers and use the remaining two fully connected layers as the classification head $c(z)$. For tabular datasets, we use a three-layer MLP (multilayer perceptron) as the classification model, where the first layer is set as the encoder and the remaining two layers are used as the classification head. Dropout is used for the first two layers with the dropout probability fixed at $0.2$. We use the same batch size of 64 for tabular datasets and 390 for the image dataset. For selecting another random sample $\{x_2, y, a_2\}$ to be neutralized with current sample $\{x_1, y, a_1\}$, we perform the selection within the current batch of training data. The hyper-parameters (e.g., learning rate and training epochs) are determined based on the model performance on the validation set, and early-stopping based on validation performance is used to avoid overfitting. The optimal temperature $T$ used to calculate the probability is set as 2.0, 5.0, 2.0 for Adult, MEPS and CelebA datasets respectively. For Eq. (3), we sample $\lambda$ from the list $[0.6, 0.7, 0.8, 0.9]$. The hyper-parameter $q$ in Eq. (5) is set as $0.2, 0.6, 0.3$ for Adult, MEPS and CelebA datasets respectively.

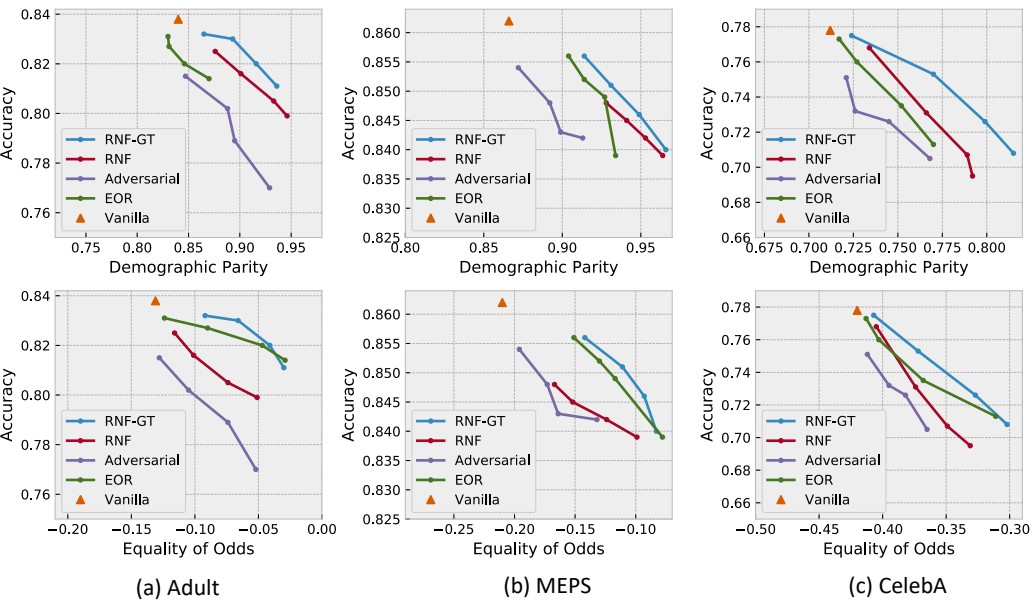

Figure 3: The fairness-accuracy curve comparison of RNF and other baselines. The first and second row depict the DP accuracy and $\Delta$EO accuracy trade-off curves, respectively. Note that there is a certain level of variance for each baseline method, and we report the average over 10 runs.

## 4.2 Mitigation Performance Analysis

We compare the mitigation performance of RNF (with proxy attribute annotations) and RNF$_{GT}$ (with ground-truth attribute annotations) with other competing methods and illustrate their fairness-accuracy curves for the three datasets in Figure 3. The hyper-parameter $\alpha$ in Eq. (4) controls the trade-off between accuracy and fairness for RNF. For Adversarial and EOR, we vary their regularization weights to obtain the corresponding performance curves. As the random seeds lead to variance in the accuracy and fairness metrics (please refer to Section E.4 in the Appendix for detailed analysis of the variability effect), we train the model for 10 times with different seeds and report the average result. Overall, we make the following key observations.

- Even though RNF does not rely on annotations for the sensitive attributes, it performs similar to baseline methods with access to such information, e.g, Adversarial training, or better than them in some cases. This makes RNF readily usable for real-world applications where protected attributes are not available in the training set.

- RNF$_{GT}$ improves mitigation performance over RNF by $10\%$ on an average across all datasets and metrics, thereby, demonstrating the benefit of using ground truth sensitive attribute annotations.

- The soft labels of RNF and RNF$_{GT}$ (obtained using a higher temperature $T$) discourage the model to assign overconfident predictions, thereby, suppressing it from capturing undesirable correlation between fairness sensitive information and class labels. Penalizing the large changes of probability as we move along the interpolation between two samples further suppresses the model from capturing the undesirable correlation.

- Direct optimization of the equality of odds metric (i.e., EOR) achieves comparable performance to RNF$_{GT}$ for all the datasets. However, it has limited improvement in terms of the demographic parity metric. Note that EOR requires instance-level annotations for the protected attributes.

- We observe that Adversarial training also performs effective mitigation by learning debiased representations. However, this happens at the expense of a higher accuracy drop for the task performance. This likely results from the loss of task relevant information while suppressing sensitive information from the representations. Additionally, we observe adversarial training to be unstable, especially for relatively complex task like image classification.

## 4.3 Classification Head Analysis

In this section, we use explainability and an auxiliary prediction task to analyze the classification head $c(z)$ for the Adult dataset. Particularly, we leverage explainability as a debugging tool to analyze the attention difference between Vanilla and RNF model with respect to the representations.

**Auxiliary Sensitive Attribute Prediction Task.** We perform a representation analysis using an auxiliary prediction task. To this end, we train another linear classifier to predict sensitive attributes using the biased representation $g(x) = z$ as input and the sensitive attribute annotations $\{a_i\}_{i=1}^N$ as the supervision signal. The linear classifier is denoted by $L_{\text{SENS}}(z) = Wz + b$, where $W$ and $b$ represent the weight matrix and bias for the linear classifier respectively. The weight matrix $W$ can be used to measure the degree of bias in each dimension of the representation $z$.

**Explanation Analysis.** We use post-hoc explainability [44] to analyze the contribution of the classification head $c(z)$. Our goal is to figure out the contribution of each dimension within the biased representation $g(x) = z$ towards the model prediction $f(x, \theta) = c(g(x))$. We train a linear classifier $L_{\text{explan}}(z) = Wz + b$ to mimic the decision boundary of the multi-layer classification head $c(z)$.

We compare the weight matrix of the two linear classifiers $L_{\text{SENS}}(z)$ and $L_{\text{explan}}(z)$ using cosine similarity. For models trained using Adult dataset, we extract the weight matrix corresponding to the protected attribute *Male* and task label *Positive*, and then we calculate the cosine similarity between the Male vector and the Positive vector. This follows from our observation in Figure 1 that the vanilla model makes use of male relevant information to make positive predictions. For the Vanilla model and RNF models listed in Figure 3 (a), we calculate the cosine similarity and report the DP-Similarity performance in Figure 4. We observe that RNF dramatically reduces the cosine similarity between positive predictions and male relevant information compared to Vanilla (from 0.272 to 0.075), by adjusting the decision boundary.

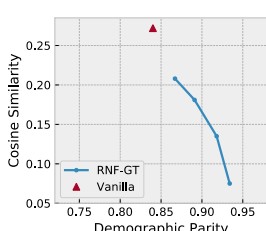

Figure 4: Analysis for the classification head.

This eventually helps the head $c(z)$ to shift its attention from fairness sensitive information to task relevant information.

## 4.4 Effectiveness of GCE Loss

We use MEPS dataset to examine the GCE loss in terms of generating proxy annotations for the protected attributes. The results are shown in Table 2. Firstly, we compare the fairness performance of GCE loss with standard CE loss (Vanilla). As expected, we observe that GCE is more biased than Vanilla in terms of the fairness metrics with 0.2% accuracy difference. Secondly, we compare RNF with several of its variants: 1) replacing GCE loss with cross entropy (CE) loss to generate proxy labels, and 2) using random annotations for the pro-

Table 2: Ablation analysis.

| Models | MEPS | | |
|---|---|---|---|
| | Accuracy | DP | $\Delta$EO |
| Vanilla | 0.862 | 0.866 | -0.210 |
| GCE | 0.860 | 0.839 | -0.249 |
| RNF-GCE | 0.839 | 0.964 | -0.099 |
| RNF-CE | 0.842 | 0.936 | -0.128 |
| RNF-Random | 0.856 | 0.902 | -0.163 |

tected attributes. For a fair comparison, except for the protected attribute annotations, we use the same set of hyper-parameters for different variants. Even though the same level of regularization is performed for the three RNF variants, RNF-GCE has a higher mitigation performance than RNF-CE with 3.0% DP improvement and only 0.3% accuracy difference, thereby, demonstrating the effectiveness of GCE in terms of separating protected groups. Another observation is that even using random annotations, our RNF framework could achieve certain level of mitigation. This is because partial samples contain different sensitive attributes, which still could serve neutralization purpose.

## 4.5 Varying Layers for the Classification Head

We use MEPS and CelebA two datasets to study the effect of an important hyper-parameter of our framework in terms of which layers to use for the encoder. Our classification model $f(x)$ contains the encoder $g(x)$ and task predictor $c(z)$. In this experiment, we vary the depth of the representation layer to examine the effect of

Table 3: Varying the layers for classification head.

| Models | MEPS | | | CelebA | | |
|---|---|---|---|---|---|---|
| | Accuracy | DP | $\Delta$EO | Accuracy | DP | $\Delta$EO |
| RNF | 0.839 | 0.964 | -0.099 | 0.668 | 0.836 | -0.290 |
| RNF-Last | 0.837 | 0.971 | -0.076 | 0.641 | 0.966 | -0.052 |

layer selection. In particular, we investigate the question: *can we debias only the very last layer, with the remaining layers as the biased encoder?* We report the results for MEPS and CelebA datasets in Table 3, where the second row depicts the result of debiasing only the last layer. On MEPS dataset, RNF-Last has better performance in terms of the two fairness metrics with only 0.2% accuracy difference. Similarly, for CelebA dataset, with 2.7% accuracy difference, RNF-Last has better fairness performance (with 13% DP improvement). This demonstrates that debiasing only the last layer can achieve similar performance compared to debiasing the last several layers.

### 4.6    Representation Neutralization with Debiased Encoder

In the discussions so far, we reported the performance of RNF while debiasing only the classification head. In this section, we analyze the impact of RNF built on top of a debiased encoder. We first use Adversarial or EOR training to learn a debiased encoder – which is subsequently used as the backbone encoder for updating the classification head using RNF. This experiment is performed on the MEPS dataset where both Adversarial and EOR methods achieve

Table 4: RNF with debiased encoder.

| Models | MEPS | | |
| --- | --- | --- | --- |
| | Accuracy | DP | $\Delta$EO |
| Vanilla | 0.862 | 0.866 | -0.210 |
| RNF | 0.839 | 0.964 | -0.099 |
| RNF_EOR | 0.834 | 0.980 | -0.049 |
| RNF_Adversarial | 0.826 | 0.971 | -0.085 |

competitive performance. We use the same hyper-parameters for different RNF variants and *report a single point in the fairness-accuracy curve*. The results are shown in Table 4. We observe RNF_EOR to achieve better fairness performance over RNF, with DP metric improvement of 1.6% and $\Delta$EO metric moves closer to 0. However, such improvement in the fairness metrics incur some loss in the task performance – where the accuracy reduces by 0.5%. We observe a similar trend with the combination of RNF and Adversarial training, where the joint combination improves the fairness metrics DP and $\Delta$EO. Similar to the previous case, this fairness improvement is achieved at the expense of some task performance degradation, where the accuracy drops by 1.3%. This indicates that our RNF is complementary to using a debiased encoder where the joint combination performs better than either of them in terms of the fairness metrics with some loss in task performance.

## 5    Conclusions and Future Work

In this work, we demonstrate that even when input representations are biased, we can still improve fairness by debiasing only the classification head of the DNN models. We introduce the RNF framework for debiasing the classification head by neutralizing training samples that have the same ground truth label but with different sensitive attribute annotations. To reduce the reliance on sensitive attribute annotations (as used in existing works), we generate proxy annotations by training a bias-intensified model and then annotating samples based on its confidence level. Experimental results indicate our RNF framework to dramatically reduce the discrimination of DNN models, without requiring access to annotations for the sensitive attributes for all the training samples. Experimental analysis further demonstrates our RNF framework to further improve in conjunction with other debiasing methods. Specifically, our RNF framework built on top of a debiased backbone encoder leads to better mitigation performance with negligible accuracy drop in the task performance. It is worth noting that our RNF framework could help alleviate the discrimination rather than eliminate it.

On the other hand, the experimental analysis indicates a mitigation gap between RNF and $\text{RNF}_{\text{GT}}$. This is because we assume zero access to the protected attribute annotations when using GCE framework. It is desirable to further boost the quality of the generated proxy annotations. In real-world applications, domain experts could be involved to annotate a small ratio of the samples for the training set. Equipped with this small ratio of high quality protected attribute annotations, we could generate proxy annotations for other training samples with a higher accuracy compared to the proxy annotations generated by GCE framework. As such, we can further boost the mitigation performance of RNF. This is a challenging topic and would be explored in our future research.

## 6    Funding Transparency Statement

The work is in part supported by NSF grants CNS-1816497, IIS-1900990, and IIS-1939716. The views and conclusions contained in this paper are those of the authors and should not be interpreted as representing any funding agencies.

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
