# A Proof of Theorem

**Proof of Theorem 1**:

Beginning with triangular inequality $|x + y| \leq |x| + |y|$ for $x, y \in \mathcal{R}$, we have Equation (2)

$$\left| \mathbb{E}_{x_i \sim P(x_i|a_i=0)} L(c(z_i), p_i) - \mathbb{E}_{x_j \sim P(x_j|a_j=1)} L(c(z_j), p_j) \right|$$
$$\leq \mathbb{E}_{x_i \sim P(x_i|a_i=0), x_j \sim P(x_j|a_j=1), |p_i-p_j| \leq \epsilon_p} \left| L(c(z_i), p_i) - L(c(z_j), p_j) \right| \tag{8}$$

For $x_1 \sim P(x_1 \mid a_i = 0)$, $z_1 = g(x_1)$ and $x_2 \sim P(x_2 \mid a_i = 1)$, $z_2 = g(x_2)$, $L(c(z_1), p_1)$ and $L(c(z_2), p_2)$ are bounded by:

$$L(c(z_1), p_2) - \epsilon_p \epsilon_L \leq L(c(z_1), p_1) \leq L(c(z_1), p_2) + \epsilon_p \epsilon_L,$$
$$L(c(z_2), p_1) - \epsilon_p \epsilon_L \leq L(c(z_2), p_2) \leq L(c(z_2), p_1) + \epsilon_p \epsilon_L, \tag{9}$$

which is specified in **Proposition 1**. Taking the upper and lower bound of $L(c(z_1), p_1)$ and $L(c(z_2), p_2)$ into $|L(c(z_1), p_1) - L(c(z_2), p_2)|$, we have

$$\left| L(c(z_1), p_1) - L(c(z_2), p_2) \right|$$
$$\leq \left| \frac{1}{2} \left( L(c(z_1), p_1) + L(c(z_1), p_2) + \epsilon_p \epsilon_L \right) - \frac{1}{2} \left( L(c(z_2), p_2) + L(c(z_2), p_1) - \epsilon_p \epsilon_L \right) \right|$$
$$\leq \left| \frac{1}{2} \left( L(c(z_1), p_1) + L(c(z_1), p_2) \right) - \frac{1}{2} \left( L(c(z_2), p_2) + L(c(z_2), p_1) \right) \right| + \epsilon_p \epsilon_L \tag{10}$$
$$= \left| L\left( c(z_1), \frac{p_1 + p_2}{2} \right) - L\left( c(z_2), \frac{p_1 + p_2}{2} \right) \right| + \epsilon_p \epsilon_L$$

To obtain the upper bound of Equation (10), we consider the second-order taylor expansion for $L(c(z_1), (p_1 + p_2)/2)$ and $L(c(z_2), (p_1 + p_2)/2)$ given by

$$L\left( c(z_1), \frac{p_1 + p_2}{2} \right) \approx L\left( \left( \frac{z_1 + z_2}{2} \right), \frac{p_1 + p_2}{2} \right) + \frac{(z_1 - z_2)^T}{2} \nabla_z L(c(z), p)|_{z=\frac{z_1+z_2}{2}, p=\frac{p_1+p_2}{2}}$$
$$+ \frac{1}{2} \frac{(z_1 - z_2)^T}{2} \nabla_z^2 L(c(z), p) \mid_{z=\frac{z_1+z_2}{2}, p=\frac{p_1+p_2}{2}} \frac{(z_1 - z_2)}{2},$$
$$L\left( c(z_2), \frac{p_1 + p_2}{2} \right) \approx L\left( \left( \frac{z_1 + z_2}{2} \right), \frac{p_1 + p_2}{2} \right) - \frac{(z_1 - z_2)^T}{2} \nabla_z L(c(z), p)|_{z=\frac{z_1+z_2}{2}, p=\frac{p_1+p_2}{2}}$$
$$+ \frac{1}{2} \frac{(z_1 - z_2)^T}{2} \nabla_z^2 L(c(z), p) \mid_{z=\frac{z_1+z_2}{2}, p=\frac{p_1+p_2}{2}} \frac{(z_1 - z_2)}{2}.$$

We relax $L(c(z_1), (p_1 + p_2)/2)$ and $L(c(z_2), (p_1 + p_2)/2)$ by its second-order taylor expansion, respectively. $|L(c(z_1), (p_1 + p_2)/2) - L(c(z_2), (p_1 + p_2)/2)|$ is bounded by

$$\left| L\left( c(z_1), \frac{p_1 + p_2}{2} \right) - L\left( c(z_2), \frac{p_1 + p_2}{2} \right) \right| = (z_1 - z_2)^T \nabla_z L(c(z), p) \mid_{z=\frac{z_1+z_2}{2}, p=\frac{p_1+p_2}{2}}$$
$$\leq ||z_1 - z_2||_2 ||\nabla_z L(c(z), p)||_2 \mid_{z=\frac{z_1+z_2}{2}, p=\frac{p_1+p_2}{2}}$$
$$\leq \lambda_z \epsilon_p \epsilon_c$$

Hence, Equation (10) is bounded by

$$\left| L(c(z_1), p_1) - L(c(z_2), p_2) \right| \leq \lambda_z \epsilon_p \epsilon_c + \epsilon_p \epsilon_L \tag{11}$$

Finally, we conclude the proof by

$$\left| \mathbb{E}_{x_i \sim P(x_i|a_i=0)} L(c(z_i), p_i) - \mathbb{E}_{x_j \sim P(x_j|a_j=1)} L(c(z_j), p_j) \right|$$
$$\leq \mathbb{E}_{x_i \sim P(x_i|a_i=0), x_j \sim P(x_j|a_j=1), |p_i-p_j| \leq \epsilon_p} \lambda_z \epsilon_p \epsilon_c + \epsilon_p \epsilon_L \tag{12}$$
$$\leq \epsilon_p (\lambda_z \epsilon_c + \epsilon_L)$$

**Proposition 1** *For $x_1 \sim P(x_1 \mid a_i = 0)$, $z_1 = g(x_1)$ and $x_2 \sim P(x_2 \mid a_i = 1)$, $z_2 = g(x_2)$, $L(c(z_1), p_1)$ and $L(c(z_2), p_2)$ are bounded by:*

$$
\begin{aligned}
L(c(z_1), p_2) - \epsilon_p \epsilon_L \leq L(c(z_1), p_1) \leq L(c(z_1), p_2) + \epsilon_p \epsilon_L, \\
L(c(z_2), p_1) - \epsilon_p \epsilon_L \leq L(c(z_2), p_2) \leq L(c(z_2), p_1) + \epsilon_p \epsilon_L,
\end{aligned}
\tag{13}
$$

*Proof*: Without the loss of generality, we only prove the bound of $L(c(z_1), p_1)$, because that of $L(c(z_2), p_2)$ can be achieved based on similar deviation. Specifically, we begin with having $L(c(z_1), p_2)$ given by:

$$
\begin{aligned}
L(c(z_1), p_2) &= L(c(z_1), p_1) + L(c(z_1), p_2) - L(c(z_1), p_1) \\
&\leq L(c(z_1), p_1) + \left| L(c(z_1), p_2) - L(c(z_1), p_1) \right| \\
L(c(z_1), p_2) &\geq L(c(z_1), p_1) - \left| L(c(z_1), p_2) - L(c(z_1), p_1) \right|
\end{aligned}
\tag{14}
$$

Note that $|p_1 - p_2| \leq \epsilon_p$ and $l(c(z_1), y = j) \leq \epsilon_L$ for $j = 0, 1$, we have the upper bound of $|L(c(z_1), p_2) - L(c(z_1), p_1)|$ given by

$$
\begin{aligned}
\left| L(c(z_1), p_2) - L(c(z_1), p_1) \right| &= \left| (p_1 - p_2)l(c(z_1), y = 0) + (p_2 - p_1)l(c(z_1), y = 1) \right| \\
&= |p_1 - p_2||l(c(z_1), y = 0) - l(c(z_1), y = 1)| \leq \epsilon_p \epsilon_L
\end{aligned}
\tag{15}
$$

Hence, $L(c(z_1), p_1)$ is bounded by the follow inequality, where that of $L(c(z_2), p_2)$ can be achieved based on similar derivation process.

$$
L(c(z_1), p_2) - \epsilon_p \epsilon_L \leq L(c(z_1), p_1) \leq L(c(z_1), p_2) + \epsilon_p \epsilon_L
\tag{16}
$$

## B  Justification for Uncertainty-based Proxy Annotation

Generating proxy annotations for sensitive attributes is based on the observation that the bias-amplified model tends to assign the privileged group more desired outcome, while assigning the under-privileged group less-desired outcome. For instance in the Adult dataset, the average prediction probability of the desired label (higher income) for the male group is 0.078 higher than that of the female group. Note that this comparison is performed for both groups with the same desired ground truth label. In contrast, the average probability of the less-desired label for the female group is 0.113 higher than that of the male group. Similarly for the MEPS dataset, the average probability of assigning the desired label to the priviledged 'white' group is 0.128 higher than that of the under-priviledged 'non-white' group; whereas the average probability of assigning the less-desired label to the 'non-white' group is 0.075 higher than that of the 'white' group. Note that these numbers are statistically significant, given that the prediction probability for each instance is within [0, 1].

## C  Benchmark Datasets

In this section, we introduce more details about the three benchmark datasets we used.

- **Adult Income Dataset** (**Adult**[3]): The main goal of this task is to predict whether a person makes over 50K a year or not. There are many protected attributes in this dataset, including gender, race, age, etc. In this work, we focus on gender bias, where the DNN models trained using standard cross entropy loss would show discrimination towards females. A sample belonging to the female group will be given much lower probability for making over 50k a year compared to a male, even if they have the same profile.

- **Medical Expenditure Dataset** (**MEPS**[4]): This is used to predict whether a person has high utilization or not. Here the utilization is determined by the total number of trips requiring some sort of medical care. We examine race bias for this dataset, where DNN models trained via cross entropy loss will be biased towards the non-white group. Here the privileged white group includes the original features RACEV2X = 1 (White) and HISPANX = 2 (non Hispanic), while the unprivileged non-white group includes all other demographic groups.

---

[3] https://archive.ics.uci.edu/ml/datasets/adult

[4] https://github.com/Trusted-AI/AIF360/blob/master/examples/tutorial_medical_expenditure.ipynb

Table 5: Comparison with several baseline methods

| | Debiasing Encoder | Debiasing Classification Head | Requiring Sensitive Attribute |
|---|---|---|---|
| Vanilla | ✗ | ✗ | ✗ |
| Adversarial | ✓ | ✗ | ✓ |
| EOR | ✓ | ✓ | ✓ |
| RNF | ✗ | ✓ | ✗ |
| RNF_GT | ✗ | ✓ | ✓ |

- **CelebFaces Attributes** (**CelebA**[5]): This is a large-scale face attributes dataset consisting of more than 200K celebrity images. Each image is labelled with 40 attribute annotations, where we only use *Male* and *Wavy_Hair* two attributes. We formulate it as a binary classification task, targeting to predict whether an image contains wavy hair or not. We focus on gender bias, where DNN models trained with standard cross entropy loss will show discrimination towards males.

## D  Comparing Baselines

In this section, we introduce more details about the comparing baselines.

- **Vanilla**: This model is trained using standard cross entropy loss. The learning rate is fixed as $10^{-3}$ for tabular datasets and $3 * 10^{-5}$ for image dataset respectively. For tabular datasets, we use the batch size of 64, and train the model for a maximum of 20 epochs. For image dataset, we use batch size of 390 and train the model for a maximum of 8 epochs. For all tasks, we use the Adam optimizer, and early stopping is used to avoid the overfitting.

- **Adversarial Training** (**Adversarial**): Consider that the classification model is $f(x) = c(g(x))$ where $g(x)$ is the encoder and $c(\cdot)$ is the classification head. For adversarial training, another adversarial classifier $c_{adv}(\cdot)$ is also constructed. The classification head $c(\cdot)$ and the adversarial classifier $c_{adv}(\cdot)$ are trained simultaneously. The goal of the classification head is to maximize the encoder's ability to predict the main classification task labels, while the goal of the adversarial classifier is to minimize the encoder's ability to predict the protected attributes. The adversarial training process is denoted as follows:

$$
\begin{aligned}
&\arg\min_{c_{adv}} \quad L(c_{adv}(g(x)), a) \\
&\arg\min_{g,c} \quad L(c(g(x)), y) - \beta_1 L(c_{adv}(g(x)), a),
\end{aligned}
\tag{17}
$$

The adversarial classifier $c_{adv}(\cdot)$ and the combination of encoder $g(x)$ and classification head $c(\cdot)$ are trained iteratively. Note that the protected attribute annotation $a$ is required in order to train the adversarial classifier $c_{adv}(\cdot)$. The hyperparameter $\beta_1$ controls the fairness-accuracy trade-off. A higher $\beta_1$ value will lead to better mitigation performance, while at the expense of lower accuracy.

- **Equalized Odds Regularization** (**EOR**): It directly optimizes the EO metric in Eq. (7):

$$
\mathcal{L}_{EOR} = \mathcal{L}_{CE} + \beta_2 \Delta EO,
\tag{18}
$$

where $\mathcal{L}_{CE}$ indicates the standard cross entropy loss, and $\Delta EO$ denotes the Equalized Odds metric. Note that the Equalized Odds metric is calculated within a training batch, and instance-level protected attribute annotations are needed to calculate the Equalized Odds metric. The hyperparameter $\beta_2$ is used to control the fairness-accuracy trade off, where a larger $\beta_2$ value will impose a stronger regularization, leading to better mitigation performance and worse accuracy.

We list the comparison between different comparing methods in Table 5, including whether the debiasing is performed at the encoder-level or at the classification-head-level, as well as whether sensitive attributes annotations are needed in the model training process.

## E  More on Experimental Analysis

### E.1  Classification Models

In this section, we introduce more details about the classification models. Since the goal of this work is not to achieve state-of-the-art prediction performance, we only use standard classification models.

---

[5] http://mmlab.ie.cuhk.edu.hk/projects/CelebA.html

- **MLP**: It contains three layers, where the dimension for both hidden layers is fixed as 50. The dimension for the input layer is 98 for Adult dataset and 138 for MEPS dataset respectively. We use Relu activation after each linear layer and also utilize the Dropout with probability of 0.2. Except Section 4.5, we use the first layer as the encoder and the rest two layers as the classification head for all other sections throughout this work.

- **CNN**: It is based on ResNet-18 [1]. Note that different from the original ResNet-18, we use two fully connected layers. The dimension for two layers is 512 and 100 respectively. Except Section 4.5, we use the convolutional layers as the encoder, and the rest two fully connected layers as the classification head for all other sections throughout this work.

### E.2 Experimental Settings

In this section, we introduce more details about the experimental settings.

- **Fairness-accuracy Curve**: In Section 4.2, we reported the fairness-accuracy trade off curve. For RNF and $RNF_{GT}$, we vary hyper-parameter $\alpha$ in Eq. (4) to draw the curve. For Adversarial and EOR, we vary the value of $\beta_1$ and $\beta_2$ respectively to get the fairness-accuracy curve.

- **A Single Point**: Besides the curve in Section 4.2, we also reported a single point in the curve as in Section 4.5 and in Section 4.6. This is obtained by fixing the hyper-parameter $\alpha$ in Eq. (4).

### E.3 Applications with Non-binary Sensitive Attributes

Our RNF can also be applied to applications with non-binary sensitive attributes. Suppose we have three groups with corresponding ground truth sensitive annotations, the loss function in Eq.(3) can be modified as follows:

$$\left| c\left( \frac{\lambda_1 z_1 + \lambda_2 z_2 + \lambda_3 z_3}{\lambda_1 + \lambda_2 + \lambda_3} \right) - c\left( 1/3 z_1 + 1/3 z_2 + 1/3 z_3 \right) \right|, \tag{19}$$

where all three lambda values are sampled from the interval [0, 1].

**Assign Proxy Annotations among Groups** Another question is that how does RNF assign proxy annotations among multiple groups using the GCE framework. A simple extension of the RNF framework to the non-binary attribute setting is to use a similar strategy to that of many multi-class classification works and treat this as one-versus-all classification task for each of the sensitive attributes. For instance, given three race groups Caucasian, African-American, and Hispanic, the GCE framework can generate proxy annotations considering the privileged Caucasian group and under-priviledged African-American and Hispanic groups. Given a sample with multiple proxy annotations corresponding to the different sensitive attributes, we can either use the soft annotations or the hard one (by argmax) to train the DNN model.

### E.4 Variability in Performance of the Models

In this section, we analyze the variability effect of fairness metrics performance in model training. DNN model training exhibits large variability due to factors like the choice of random seeds, training/validation/test splits, data loading order, batch size, optimizer, learning rate, training epochs, etc. We report the performance of RNF over 5 runs with different seeds along with standard deviation for the Adult dataset. Note that we use the same train/test/valid split for our model and the results correspond to a single point on the fairness accuracy curve. Please refer to Section 4.1.2 for hyper-parameters and other experimental settings. The

Table 6: Variability in performance.

| Models | MEPS | | |
|---|---|---|---|
| | Accuracy | DP | $\Delta$EO |
| Run 1 | 0.817 | 0.918 | -0.041 |
| Run 2 | 0.817 | 0.931 | -0.048 |
| Run 3 | 0.808 | 0.928 | -0.056 |
| Run 4 | 0.811 | 0.921 | -0.051 |
| Run 5 | 0.819 | 0.917 | -0.049 |
| Mean | 0.814 | 0.923 | -0.049 |
| Standard deviation | 0.004 | 0.006 | 0.005 |

performance with 5 runs and the corresponding mean and standard deviation are given in Table 6. We observe that there is variability effect for the fairness metrics performance when using different random seeds. However, the performance of RNF is relatively consistent across different runs with a very small standard deviation.