# OpenReview forum: "Fairness via Representation Neutralization"
_NeurIPS.cc/2021/Conference — NeurIPS 2021 Poster_

### Official Review · Reviewer_Ho2y · 2021-06-30

**Rating:** 6
**Confidence:** 4

**Summary:**

This paper proposes an unfairness mitigation technique for DNN models called Representation Neutralization for Fairness (RNF), which debiases the classification head without touching the possibly-biased representations. The idea is to first train a teacher network for the representations and then a representation neutralization network that trains on neutralized features (random combinations of samples with the same label, but different sensitive attribute values). Another contribution is to generate proxy annotations for sensitive attributes in case they are not available. Experiments show that RNF outperforms several baselines in terms of accuracy and fairness.

**Ethical Concerns:**

I do not see any ethical concerns.

**Limitations And Societal Impact:**

The limitations of the work are sufficiently discussed in the supplementary, but should probably appear in the body of the paper. There is no discussion on potential negative societal impacts. One possible discussion is whether optimizing fairness may reduce accuracy too much, which can be viewed as a negative outcome.


**Main Review:**

Overall, the paper proposes an intriguing approach for unfairness mitigation by only debiasing the classification head. The idea is simple, yet effective. However, some critical comparisons with other methods seem to be missing.

Strong points:

* Simple and effective approach for debiasing without touching the representations.

* Adapting Mixup techniques for generating neutralized features is a neat idea.

* The proposed method can be combined with any debiased model to further improve fairness.

* Experimental results look promising on multiple datasets and prominent fairness measures, even when annotations for sensitive attributes are not available.

* The paper is well written.

Weak points:

* During representation neutralization, mixing random samples with different sensitive attributes seems rather ad-hoc. Have the authors tried limiting the samples to mix, say by distance? If that does not work, why?

* The key assumption for generating proxy annotations is questionable. A biased model may be over-confident for any subgroup of the data, and there is no guarantee this happens to be one of the sensitive groups. The authors should experimentally verify that the proxy annotations really align with sensitive attribute annotations. That is, do the samples with proxy annotations 0 (1) really have gender male (female)?

* Some critical comparisons seems to be missing in the experiments.

     - RNF seems very close to post-processing unfairness mitigation techniques as it can be viewed as neutralization (post-processing) after training the teacher network. Hence, there should be an experimental comparison with Hardt's method, which the authors already cite ([27]).

     - The authors cite the relevant work Fair mixup [40], but do not make any comparisons with it either.

     - Since the authors compare with EOR (direct optimization of equalized odds), there should also be a comparison with Zafar's method, which optimizes demographic parity. Zafar et al., "Fairness Constraints: Mechanisms for Fair Classification", AISTATS 2017.

* Although the authors claim that the proposed method can be extended to non-binary sensitive attributes, some clarifications would be helpful.

     - When generating proxy annotations, one demographic group has over-confident predictions, and the alternative group has less-confident ones. If there are more than two sensitive groups, how does RNF assign proxy annotations among groups?

     - Smoothening neutralization: In Eq. (3), how can we assign lambda values to multiple sensitive groups?

* Optionally, it would be nice to also use the ProPublica COMPAS dataset in the experiments.

======

The author response addresses my main concerns above. Thanks for the clarifications and new experiments.


**Time Spent Reviewing:**

4

---

> ### Comment · Reviewer_Pdht · 2021-08-07
> **Similar question regarding proxy annotations**
>
> I am one of the other reviewers and had a similar question about proxy annotations. Thanks for raising it. In my review, I write:
>
>
> > "You say, "A biased model generates over-confident predictions for one demographic group, while giving much lower scores for the alternative group." Can you justify this claim? Or is this true by definition? What happens if you apply your technique to an unbiased model?"

---

> ### Author Response · Authors · 2021-08-10
> **Response to Reviewer Ho2y’s Comments**
>
> We thank the reviewer for your positive feedback and insightful suggestions.
>
> **1. Leveraging distance for selecting samples for neutralization.**
>
> Thanks for this suggestion. Our work is motivated by the Mixup works [20, 21] that demonstrate robust model performance for generic image classification tasks. In this work, we focus on a different problem setting to leverage such techniques for improving fairness-utility trade-off of neural models -- where the core idea is to mix representations of samples with different sensitive attributes. Thereafter, we propose techniques that do not need access to the sensitive attribute information.
>
> In the above process, we randomly select samples (with a given sensitive attribute) to interpolate hidden state representations. This form of Manifold Mixup has been shown to flatten class-specific representations (reducing number of directions with significant variance) such that a randomly sampled hidden representation within the convex hull spanned by the data in this space is more likely to have a classification score with lower confidence (higher entropy) [21]. We exploit this form of regularization to provide better fairness-utility trade-off. We will add these clarifications for our choices and connection to prior work in our revised manuscript. Furthermore, we can explore better sample selection strategies by exploring sample diversity and model uncertainty that we will discuss in future works.
>
>
> **2 Justification for the assumptions and effectiveness of generating proxy annotations.**
>
> Yes, the biased model could give over-confident predictions for any subgroup of data, but for different labels. Particularly, the bias-amplified model tends to assign the privileged group more desired outcome, while assigning the under-privileged group less-desired outcome. For instance in the Adult dataset, the average prediction probability of the desired label (higher income) for the male group is 0.078 higher than that of the female group. Note that this comparison is performed for both groups with the same desired ground truth label. In contrast, the average probability of the less-desired label for the female group is 0.113 higher than that of the male group. Similarly for the MEPS dataset, the average probability of assigning the desired label to the priviledged ‘white’ group is 0.128 than that of the under-priviledged ‘non-white’ group; whereas the average probability of assigning the less-desired label to the ‘non-white’ group is 0.075 higher than that of the ‘white’ group. Note that these numbers are statistically significant, given that the prediction probability for each instance is within [0, 1].
>
> Our proxy annotation can be regarded as a special case of using distance based metric to select mixing samples (see the previous question) for regularization. For example, if the current example has an over-confident prediction for the desired label (i.e., larger than 0.993 confidence), we will select samples with the same ground truth label but with less confident predictions (less than 0.993). This is equivalent to selecting samples with a large distance in the softmax probability space of the bias-amplified network $f_B(x)$.
>
> To verify the effectiveness of proxy annotations, we calculate their alignment with ground truth sensitive attributes, using the metrics of precision, recall and F-Measure with results as follows:
>
>
> |Dataset| Precision        | Recall           | F-Measure  |
> | -------------| ------------- |:-------------:| -----:|
> |Adult| 0.647      | 0.592 | 0.604 |
> |MEPS| 0.651      | 0.643      |   0.646 |
>
>
> We further validate the effectiveness of the GCE framework in Table 2, where RNF-GCE (with bias-amplified proxy annotations) has better performance than RNF-Random (with random sensitive attribute annotations). However, such proxy annotations, as noted by the reviewers, are still noisy and we observe a performance gap between them and the ground truth annotations. This performance gap is illustrated in Figure 3 between the RNF and RNF-GT (described in lines 255-256). We also mention the limitation of the proxy annotations in Section E at Appendix (line 635-645).
>
> We leverage proxy annotations to train the GCE framework since we assume zero access to the ground truth annotations for the protected attributes. In real-world applications, we can leverage domain experts to annotate a small amount of data for the sensitive attributes (say, 5%) that can be used to train our model to generate better proxy annotations in contrast to the completely unsupervised GCE setup.
>
>
> **3. Some comparisons are missing in the experiments.**
>
> We conduct experiments by comparing with the baseline that directly optimizes the Demographic Parity metric as recommended by the reviewer. For implementation details, please refer to lines 604-609 in Appendix. Different from Equation 18 (Appendix), we regularize the Demographic Parity metric to encourage it to be close to 1. The results for the Adult dataset are given below.
>
> |Models| Accuracy        | DP           | $\Delta$EO  |
> | -------------| ------------- |:-------------:| -----:|
> |Vanilla| 0.838      | 0.840 | -0.131 |
> |DP baseline | 0.830      | 0.889      |   -0.109 |
> |RNF (ours)| 0.831      | 0.896      |   -0.089 |
>
> The first row denotes results for the vanilla model trained using cross entropy loss, whereas the second row is the result for optimizing Demographic Parity metric. The last row is the result for the proposed method which indicates that our RNF method can achieve better fairness metric scores when the models have similar accuracy performance.
>
>
> **4. Non-binary sensitive attributes: how does RNF assign proxy annotations among groups?**
>
> A simple extension of the RNF framework to the non-binary attribute setting is to use a similar strategy to that of many multi-class classification works and treat this as one-versus-all classification task for each of the sensitive attributes. For instance, given three race groups Caucasian, African-American, and Hispanic, the GCE framework can generate proxy annotations considering the privileged Caucasian group and under-priviledged African-American and Hispanic groups. Given a sample with multiple proxy annotations corresponding to the different sensitive attributes, we can either use the soft annotations or the hard one (by argmax) to train the DNN model.
>
> **5. Non-binary sensitive attributes: How can we assign lambda values to multiple groups?**
>
> Suppose we have three groups with corresponding ground truth sensitive annotations, the loss function in Equation 3 can be modified as follows:
> $|c(\frac{\lambda_1 z_1 + \lambda_2 z_2 +\lambda_3 z_3}{\lambda_1+\lambda_2+\lambda_3}) - c(⅓ z_1 + ⅓ z_2 +⅓ z_3)|$,
>
> where all three lambda values are sampled from the interval [0, 1].
>
> **6. Limitations and societal impacts.**
>
> Thanks for these suggestions. In the revised manuscript, we will move the limitations of the work from Appendix to the main body of the paper. We will also discuss the negative impacts of task performance drop when overly optimizing for fairness metrics, although in this work we strive to obtain a good fairness-utility trade-off.

---

### Official Review · Reviewer_FmVc · 2021-07-14

**Rating:** 7
**Confidence:** 4

**Summary:**

The authors proposed a new technique to reduce the discrimination of DNN models by debiasing only the task-specific classification head of DNN models  (termed as Representation Neutralization for Fairness, RNF). Furthermore, to handle the cases where sensitive attribute annotations are not available, they leveraged a bias-amplified model to generate approximate annotations for sensitive attributes. They showed that the RNF effectively reduces the bias of DNN models with a minimal decrease in model performance.

**Limitations And Societal Impact:**

There's no foreseeable negative societal impact of this work.

**Main Review:**

The work is innovative as previous work on this topic mostly focused on learning debiased encoders, whereas the current approach focuses on learning debiased decoders. Also, existing methods require annotations of the sensitive attributes but the current method removes this constraint as it can generate the proxy sensitive attribute via a biased-amplification approach.

 The submission is technically sound. The authors not only demonstrated that their approach mitigates bias (3.2) but also conducted thorough ablation studies to confirm the necessity of different model components (3.4). The analysis of the classification head further illuminates the effect of debiasing the classification head. The author found that RNF dramatically reduced the cosine similarity between positive predictions and sensitive attributes (e.g. male) compared to the Vanilla model by adjusting the decision boundary. I do have a question related to the results: since there's a tradeoff between accuracy and fairness, I wonder if the authors could compute a single metric combining accuracy and fairness (e.g., just like how f1-score takes account of both precision and recall) so that we could compare multiple models more directly.

The submission is clearly written and well organized. The results are important in the sense that it works complementary to the debiased encoder. When combined with debiased encoders, the model performs better than either of them alone in terms of the fairness metrics albeit with some loss in task performance. The authors may think more about how to improve fairness without the performance drop.

-----------

Updated: I've read the authors' responses to my review as well as the ones to the other reviewers'. I think their responses addressed my questions. I appreciate the clarifications on the single metric and the trade-off between fairness and performance. I will keep my score unchanged as it reflects my best judgment of the work.

**Time Spent Reviewing:**

10 hours

---

> ### Author Response · Authors · 2021-08-10
> **Response to Reviewer FmVc’s Comments**
>
> We thank the reviewer for the encouraging words and helpful comments.
>
> **1 A single metric combining accuracy and fairness.**
>
> For a single unifying metric, we can use the Area Under the Curve (AUC) score. The higher the AUC, the better the performance of the model at achieving accuracy and fairness trade-off. Note that, we use the AUC score on the validation set for determining the splitting threshold for proxy annotations for sensitive attributes.
>
> **2 Think about how to improve fairness without the performance drop.**
>
> Thanks for this suggestion. Improvement in the fairness metrics such as demographic parity and equality of opportunity usually comes at the expense of task-specific performance drop. However we still can optimize for the fairness and utility trade-off. As mentioned in the last question, a desirable mitigation solution is to have a high AUC score. Exploring mitigation solutions with even less performance drop is a challenging research problem that we would like to explore in future work.

---

### Official Review · Reviewer_Pdht · 2021-07-16

**Rating:** 7
**Confidence:** 3

**Summary:**

This paper presents a method for fine-tuning the upper layers of a DNN (the "classification head") with the goal that it becomes insensitive to information in the encoded representation that correlates with a protected attribute. The technical contribution leverages the *mixup* approach to present "neutralized" encodings to the classification head, which learns to project them to softened target probabilities. Another key contribution is that this process can be done *without* access to ground truth labelling of the protected attribute. As a proxy, the authors assume that the unneutralized model will predicatably generate biased output in favour of a specified demographic group. They use that output to generate proxy annotations to take the place of ground truth labelling of the protected attribute.

The work is generally well-written and I have many minor comments, questions, and some suggestions for improvement and clarification throughout.

~However, I have concerns about the evaluation. These concerns are the primary basis for my ultimate recommendation to not accept. If these concerns are addressed or if I have missed something, I am open to adjusting my recommendation.~

The author(s) have provided very helpful responses:

* they "will rephrase the claim that 'RNF… outperform baseline methods…' in the revised manuscript to instead highlight the resource benefits of RNF over the baseline methods with similar or better performance in some cases,"
* while not going as far as statistical significance tests, they have provided more information that allows me to better assess whether the observations are mere noise or true effects; they will average across more (10) runs and will include error bars in the figure
* they have agreed to re-organize some of the presentation to make more clear some of the points I had questions about

Based on these commitments, I have revised my recommendation to "7: good paper, accept."

**Limitations And Societal Impact:**

Authors could point out that some anti-discrimination regimes require the *elimination* not the mere *reduction* of disparate outcomes in government decision-making.

**Main Review:**

# Originality

The proposed method seems novel and parts of it surprise me! The work seems to adequately cite its motivations and cites what it builds off of.

I was most surprised by the claim that the framework "does not require any access to the protected attributes during training."

# Quality & Clarity

**I wonder throughout whether the phrase "representation neutralization" is a good name for this technique.** You admit that you are debiasing *only* the classification head and accept that the encoded representation itself *may be biased*. And you position your work in *contrast* to existing methods that learn debiased representations. I do see how you "neutralize" the representation as part of fine-tuning the classification head, but at inference time, the representation reaching the classification head is not neutralized in the same sense. Instead, the classification head has just hopefully become insensitive to the information correlated with the protected attribute. I don't know if I have a good suggestion for an alternative name. Maybe, something like desensitized classification. In any case, it's just a name, but it did put the wrong idea in my head for the early parts of the paper.

## Introduction

You seem to be alluding to COMPAS at line 25. I think works frequently cite this ProPublica article when referring to COMPAS: https://www.propublica.org/article/machine-bias-risk-assessments-in-criminal-sentencing

I also would have expected citations to Calo & Citron, "The Automated Administrative State" (https://scholarlycommons.law.emory.edu/elj/vol70/iss4/1/), and Kate Crawford, *Atlas of AI*.

Suggest citing to Edwards & Storkey, "Censoring Representations with an Adversary" (ICLR, 2016).

At line 23, you refer to *the* recidivism algorithm. This implies that there is only one. Suggest something like: "COMPAS, an algorithmic recidivism predictor, is likely to associate..."

At line 34: "The suppression of fairness sensitive information might also remove useful information that is task relevant." Much work in this area acknowledges that this is the trade-off that anti-discrimination regimes require: even "rational discrimination" is not okay. See Prince & Schwarcz, "Proxy Discrimination in the Age of Artificial Intelligence and Big Data" (https://ilr.law.uiowa.edu/print/volume-105-issue-3/proxy-discrimination-in-the-age-of-artificial-intelligence-and-big-data).

I wonder whether it is helpful to refer to the correlation with the sensitive attribute as "spurious." These correlations are often the result of embedded social structures. The correlation is not spurious in the sense that it arose from randomness in a particular sample. Rather, the correlation has arisen *for real* in the population because of our decisions as society.

## Representation Neutralization for Fairness

Throughout this paper, until I got to 3.1.2, I was very curious and confused about how you determined what the "classification head" was. This confusion arises because of the way you talk about it in the early sections. You refer to "the task-specific classification head of a DNN" as if it is a clearly identified add-on. In actuality, it is just the top layer(s) of the DNN that you have chosen to set aside for fine-tuning. If there is any way you can clarify this early on, that would be helpful.

At 76, you say you consider "binary sensitive attributes" but go on to choose gender for the examples. Consider adding a caveat (even just a footnote) that gender is only binary as coded in the data, despite it not being a binary in reality.

## Analysis of the Classification Head

Again, it would be helpful to note that there is in general no such thing as a distinct "classification head" other than how you have decided to fragement your network(s) for analysis / fine-tuning purposes.

At line 101, you distinguish between "useful task relevant classification information" and "bias information from protected attributes." But this isn't a true disctinction and that is part of the problem. The issue is that the protected attributes have *become correlated* with the relevant classification target and thus *become* useful task relevant classification information. The goal of anti-discrimination frameworks is to require decision-makers to ignore that correlation *even if* it would improve some performance metric.

## Representation Neutralization for Debiasing Classification Head

Again, you make the distinction between "sensitive attributes" and "task-relevant information." The problem is that sensitive attributes have *become* task-relevant information. Perhaps you mean to distinguish between "sensitive attributes" and "permitted factors"?

Here, it becomes clear that you freeze the encoder *after* it has learned what you are calling a spurious correlation, which you hope that your fine-tuned classification head can ultimately ignore.

A question about lines 139-141. You say, "From the input perspective, the neutralization of representations suppresses the model from capturing the spurious correlation between fairness sensitive information in the representation with the class labels." Were you being careful here in saying "suppresses" rather than "prevents"? If so, that matches how I understand this would work. I don't see how your smoothing/averaging would necessarily eliminate the spurious correlation from making it through, but could suppress it to an extent. Am I misunderstanding this? Regardless, this wasn't critical to my review, but just a place where you could perhaps be clearer if I did misunderstand you.

I think it would be helpful to emphasize early on that the neutralization only happens during the fine-tuning of the hopefully debiased classification head. At inference time, the features just come in as encoded, to a classification head that has hopefully learned to not exploit any of the information correlated with the sensitive attribute.

The **smoothing neutralization** seems to be introducing a penalty for large changes in soft-max probabilities as you move along the interpolation between samples. Is this correct?

## Generating Proxy Annotations for Sensitive Attributes

You say, "A biased model generates over-confident predictions for one demographic group, while giving much lower scores for the alternative group." Can you justify this claim? Or is this true by definition? What happens if you apply your technique to an unbiased model?

I think the phrase "over-confident" is adding some confusion here. You don't mean merely that predictions are close to 1 or 0, but rather, that there is a demographic group that tends to get the less-desired outcome and another demographic group that tends to get the more desired outcome.

Ther paragraph from lines 186--202 is very dense and some aspects come out of nowhere. The "splitting threshold" for example. How is it used? How do you use the fairness-accuracy tradeoff to determine its value? It would be helpful if you took more space to explain this all.

## Experiments

## Experimental Setup

## Fairness Metrics, Benchmark Datasets, and Baselines

Generally very clear.

At line 217, you say "bias towards the female group." This is ambiguous, as it could mean bias in favour of female groups or bias disfavouring the female group. An alternative: "vanilla trained models depict bias that predicts females earn less."

At line 220, you say "discrimination towards." This is less ambiguous, so you could use this at 217 too.

## Implementation details

Very clear, other than the purpose of the thresholds at line 243 (as I mentioned above was also unclear at line 196).

Your description here of how you split the networks into representation layer and classification head is something that would have been helpful much earier. Maybe not this full amount of detail, but just that it's a somewhat arbitrary decision and nothing inherent about the division.

## Mitigation performance analysis

**This is the section that gives me the most concerns about this paper.**

You claim that "RNF... outperforms baseline methods..." However, these figures (Figure 3) do not convince me of this. The datapoints are averages across three runs. But, just looking at how non-smooth the fairness-accuracy curves are, and how close these curves are to each other, I am not sure averaging across three runs was sufficient to be able to be confident we are not observing noise from a small set of experimental runs. In Figure 3(a), under equality-of-odds analysis, EOR might beat RNF, and RNF-Gt looks similar to EOR. In Figure 3(b), under demographic-parity analysis, RNF and RNF-GT look similar to EOR. In Figure 3(b), under equality-of-odds analysis, EOR might be beating RNF, EOR looks similar to RNF-GT, and RNF looks similar to adversarial. Figure 3(c), under demographic-parity analysis, this figure does look like it is consistent with your conclusions. Figure 3(c), under equality-of-odds analysis, EOR may be better than RNF, EOR might be similar to RNF-GT. Regardless, it is hard for me to draw conclusions from this small set of 3-run experiments.

## Classification Head Analysis

This is a creative way to demonstrate correlation between fairness-sensitive information and information used by the classification head.

## Hyperparameter and ablation analysis

I was really hoping you would assess the effect of where you draw the line between encoder and classification head. Great to see this analysis here.

## Representation Neutralization with Debiased Encoder

This reports only a single point on the fairness-accuracy curve. Were these also averages across three runs? A single point on the curve isn't enough to be able to draw a conclusion.

# Revised Recommendation

The author(s) have provided very helpful responses:

* they "will rephrase the claim that 'RNF… outperform baseline methods…' in the revised manuscript to instead highlight the resource benefits of RNF over the baseline methods with similar or better performance in some cases,"
* while not going as far as statistical significance tests, they have provided more information that allows me to better assess whether the observations are mere noise or true effects; they will average across more (10) runs and will include error bars in the figure
* they have agreed to re-organize some of the presentation to make more clear some of the points I had questions about

Based on these commitments, I have revised my recommendation to "7: good paper, accept."

**Time Spent Reviewing:**

12

---

> ### Comment · Reviewer_Pdht · 2021-08-07
> **To clarify my position regarding evaluation**
>
> I recognize that the other reviewers have not had the same concern I do regarding the robustness of the evaluation. Since this is the primary reason for my recommendation to reject, I would like to explain what would change my position on this point.
>
> I am familliar with how much variability generally exists between experimental runs due to random seeds, different training/test splits, etc. when training deep neural networks. An average of three runs just doesn't seem to rule out that what we are seeing is noise.
>
> We need to be able to distinguish between 1) are we seeing noise? and 2) is what we're observing because of true differences in the performance of the models?
>
> I could be convinced by statistical tests or maybe even error bars.

---

> > ### Author Response · Authors · 2021-08-10
> > **Response to Reviewer Pdht’s Comments**
> >
> > We thank the reviewer for the thoughtful comments and discussion. We particularly thank the reviewer for the two clarification reviews. We first answer the two major concerns in terms of robust performance comparison and proxy annotations, and then reply to the other questions and concerns.
> >
> > **1. Mitigation performance analysis.**
> >
> > ***1.1. Mitigation performance comparison between RNF, EOR and Adversarial Training.***
> >
> > Firstly, we would like to point out that EOR and Adversarial training require annotations for the protected attributes (line 263), whereas RNF does not require any annotation for the sensitive attributes. Furthermore, RNF outperforms such baselines that require additional sensitive annotation in some cases. We will rephrase the claim that “RNF… outperform baseline methods…” in the revised manuscript to instead highlight the resource benefits of RNF over the baseline methods with similar or better performance in some cases. Secondly, since the EOR baseline directly optimizes the equality of opportunity metric (details given in line 604-609 at Appendix), it achieves strong performance on that metric (see the second row of Figure 3). However, its demographic parity performance is not as competitive as the equality of opportunity metric.
> >
> > ***1.2. Variability in performance of the models.***
> >
> > We agree with the reviewer that DNN model training exhibits large variability due to factors like the choice of random seeds, training/validation/test splits, data loading order, batch size, optimizer, learning rate, training epochs, etc. To alleviate the reviewer's concern, we further report the performance of RNF over 5 runs with different seeds along with standard deviation (in contrast to the 3 runs used in our submission) for the Adult dataset.  Note that we use the same train/test/valid split for our model and the results correspond to a single point on the fairness accuracy curve. Please refer to Section 3.1.2 for hyper-parameters and other experimental settings.
> >
> >
> >
> > |Runs| Accuracy        | DP           | $\Delta$EO  |
> > | -------------| ------------- |:-------------:| -----:|
> > |Run 1| 0.817      | 0.918 | -0.041 |
> > |Run 2| 0.817      | 0.931      |   -0.048 |
> > |Run 3| 0.808      | 0.928      |   -0.056 |
> > |Run 4| 0.811      | 0.921      |   -0.051 |
> > |Run 5| 0.819      | 0.917      |   -0.049 |
> > |(mean, std)| (0.814,0.004) | (0.923,0.006) |(-0.049, 0.005)|
> >
> > From the above table, we observe that the performance of RNF is consistent across different runs with a very small standard deviation. According to the reviewer recommendation, we will update Figure 3 in the revised version using the average of 10 runs and also analyze the influence of different factors (e.g, random seeds, batch size, training epochs, etc) for the fairness performance in Appendix.
> >
> > **2. Clarification about the proxy annotations.**
> >
> > ***2.1. ‘Over-confident’ predictions from biased models.***
> >
> > Yes, the understanding is correct. The bias-amplified model tends to assign the privileged group more desired outcome, while assigning the under-privileged group less-desired outcome. For the Adult dataset, we report the average results comparison for the training set (averaged for 5 runs). The average prediction probability of the desired label (higher income) for the male group is 0.078 higher than that of the female group. Note that this comparison is performed for both groups with the same desired ground truth label. In contrast, the average probability of the less-desired label for the female group is 0.113 higher than that of the male group. Similarly for the MEPS dataset, the average probability of assigning the desired label to the priviledged ‘white’ group is 0.128 higher than that of the under-priviledged ‘non-white’ group; whereas the average probability of assigning the less-desired label to the ‘non-white’ group is 0.075 higher than that of the ‘white’ group. Note that these numbers are statistically significant, given that the prediction probability for each instance is within [0, 1]. We report additional experimental results and alignment with ground-truth labels in the response for Reviewer Ho2y.
> >
> > ***2.2. What happens for an unbiased model?***
> >
> > The probability gap between two protected groups is much lower. Take the 5 runs for Adult dataset in question 1.2 for example. The average prediction probability of the desired label (higher income) for the male group is 0.012 higher than that of the female group, much lower compared to the 0.078 by GCE model. The average probability of the less-desired label for the female group is 0.049 higher than that of the male group, much lower compared to the 0.113 by GCE model. This can also be reflected from the improved fairness performance (e.g., demographic parity and equality of opportunity metrics) of the mitigated models.
> >
> >
> > ***2.3. Selection of the splitting threshold.***
> >
> > Considering the Adult dataset as an example, we randomly sample 20% of the data from the training set. We generate proxy annotations for this sampled training dataset to train our RNF model. Based on the fairness-accuracy performance of the model on the validation dataset, we select the splitting threshold corresponding to the best AUC score. This is based on the observation that with better proxy annotations, there is less trade-off between the fairness and accuracy metrics and thus a higher AUC score (see the performance gap between RNF and RNF-GT in Figure 3). After selecting the splitting threshold, we generate the proxy annotations for the whole training dataset. We will add these details in the revised manuscript.
> >
> >
> > **3. Whether "representation neutralization" is a good name?**
> >
> > We perform representation neutralization only during training, and not during inference. Thanks to the reviewer for suggesting alternative names. In the revised manuscript, we are considering renaming the mitigation solution to Debiasing Classification Head via Representation Neutralization Training to emphasize that neutralization is merely performed during training for only debiasing the classification head, so as to reduce confusion for readers.
> >
> > **4. References for COMPAS and others.**
> >
> > We will add the suggested references in the updated manuscript.
> >
> > **5. Changing the description of “spurious correlation”.**
> >
> > Thanks for this suggestion. We agree that the correlations are typically the result of embedded social structures, and would like to change its name from “spurious correlation” to “undesirable correlation”.
> >
> > **6. Clarification of the representation layer and classification head.**
> >
> > We defined our problem setup and semantics of the classification head in the third paragraph of Section 2 (lines 78-83). We will emphasize the description in the revised manuscript so as to reduce confusion for the readers.
> >
> > **7. Binary sensitive attributes.**
> >
> > Thanks for the great suggestion. Gender is not binary in reality as there are many different gender identities, such as male, female, transgender, to name a few. In this work, we consider gender as a binary sensitive attribute due to the limitation of the benchmark dataset that encodes gender as a binary variable. We will add a footnote to emphasize this point in the updated manuscript.
> >
> > **8. Suppressing or preventing the undesirable correlation?**
> >
> > Yes, the understanding is correct. We aim to suppress the classification head from capturing the undesirable correlation between fairness sensitive information with certain class labels, rather than preventing or eliminating the correlation.
> >
> > **9. Smoothing neutralization.**
> >
> > Yes, we aim to penalize the large changes in softmax probability when we move along the interpolation between two samples.
> >
> > **10. ‘bias towards’ and ‘discrimiantion towards’.**
> >
> > Yes, we are expressing that the vanilla model tends to show ‘discrimination towards’ females since ‘vanilla trained models depict bias that predicts females earn less’.
> >
> > **11. Representation neutralization with debiased encoder.**
> >
> > Yes, the results listed in Table 4 are average over three runs. Due to the 9 page limit for the main document, we only reported results for a single point in the curve.
> >
> > **12. Limitations and societal impact.**
> >
> > We acknowledge that in the government decision-making regime, it requires the elimination of the disparate rather than just reduction. We will point out that our RNF framework could help alleviate the disparate impact rather than eliminate it.

---

> > > ### Comment · Reviewer_Pdht · 2021-08-14
> > > **Follow-up questions**
> > >
> > > Thank you for your thorough reply and working to address my reservations. I have a couple of follow-up questions.
> > >
> > > 1. In Tables 2, 3, 4, where you report results for a single point on the curve, how do you choose where on the curve you report the $\text{DP}$ or $\Delta\text{EO}$. It looks like maybe you've tried to pick points on the curve with similar accuracy, but they're not exactly identical. Can you explain how you chose those particular points to report?
> > >
> > > 2. You mention that "we will update Figure 3 in the revised version using the average of 10 runs." I see that the standard deviations are quite tight. Would these show up well as error bars in the figure? Or maybe the error bars would be too tiny or make the figure too cluttered. Reporting those standard deviations somehow, even if just in the text or figure capture, would really help readers to understand that you've likely observed a real effect here.
> > >
> > > In summary, subject to your responses to the above questions, and given that you "will rephrase the claim that 'RNF… outperform baseline methods…' in the revised manuscript to instead highlight the resource benefits of RNF over the baseline methods with similar or better performance in some cases," I expect I will be able to revise my recommendation.

---

> > > > ### Author Response · Authors · 2021-08-16
> > > > **Response to the Follow-up Questions**
> > > >
> > > > We thank the reviewer for the follow-up questions and helpful discussions.
> > > >
> > > > **1. Tables 2, 3, 4 report results for a single point on the curve. How did you choose those particular points to report?**
> > > >
> > > > Tables 2, 3, and 4 essentially compare the mitigation performance for different RNF variants.  For a fair comparison, we use the same set of hyper-parameters for different RNF variants. In particular, $\alpha$ in Equation (4) is used for controlling the fairness and accuracy trade-off in the RNF framework and thus for drawing the fairness-accuracy curve. For all three tables, we fix the regularization weight as $\alpha = 0.2$ to perform the same level of regularization for different RNF variants and report the corresponding numbers in the fairness-accuracy curve. Note that $\alpha = 0.2$ corresponds to the bottom right corner point of the RNF curve in Figure 3 (b). We discussed the above in lines 307-312 on Section 3.4, and in lines 336-337 on Section 3.5. Please note that the other baselines Vanilla and GCE in Tables 2 and 4 do not involve any regularization or trade-off. We would like to emphasize this point in the revised manuscript to make it more clear to the readers.
> > > >
> > > >
> > > > **2. Reporting those standard deviations somehow, even if just in the text or figure capture, would really help readers to understand that you've likely observed a real effect here.**
> > > >
> > > > Thanks for this nice suggestion. We agree with the reviewer that it is desirable to report both the average and the standard deviation across multiple runs. Firstly, we will organize our response to question 1.2 (i.e., variability in performance of the models.) into a separate subsection in the revised manuscript. In this subsection, we will report the fairness accuracy curve with error bars (as a separate figure), to make readers better understand the variability effect of fairness metric performance resulting from factors like choices of random seeds. Secondly, in terms of Figure 3, we are considering reporting the average over 10 runs to prevent overloading Figure 3 with overwhelming information. In the caption of Figure 3 as well in the text, we will emphasize that there is a certain level of variance for each baseline method, to make the readers aware of the variability effect and link to the subsection and separate figure as outlined in the previous point.

---

### Official Review · Reviewer_Aa1C · 2021-07-18

**Rating:** 7
**Confidence:** 3

**Summary:**

The authors propose a mitigation technique, namely, Representation Neutralization for Fairness (RNF) that achieves fairness by debiasing only the task-specific classification head of DNN models. They leverage samples with the same ground-truth label but different sensitive attributes and use their neutralized representations to train the classification head of the DNN model. Experimental results over several benchmark datasets demonstrate our RNF framework to effectively reduce discrimination of DNN models with minimal degradation in task-specific performance.

**Limitations And Societal Impact:**

The authors adequately addressed the limitations.

**Main Review:**

1. The paper has dealt with the problem of bias mitigation methods for DNN models which is a very common, prevailing and still challenging problem.
2. The model is well described with proper model figure, dataset description, problem statement, adequate references.
3. The algorithm is clear. It provides reasonably complete information that will help reproduce the results, including the methodology, pseudo-code, and empirical evaluation.
4. The paper is accompanied by an adequate set of experiments for evaluating the effectiveness of the solutions the authors propose.

**Time Spent Reviewing:**

4

---

> ### Author Response · Authors · 2021-08-10
> **Response to Reviewer Aa1C’s Comments**
>
> We thank the reviewer for the comprehensive summarization of our work and for providing these constructive and insightful comments.
>
> Paper6943 authors

---

### Author Response · Authors · 2021-08-10
**General Comments for All Reviewers**

We thank all the reviewers for providing many constructive comments and helpful feedback. We are encouraged to find that they have found our contributions to be novel (Pdht, FmVc), well-written (Pdht, Ho2y, Aa1C), well-evaluated (FmVc, Aa1C) and technically sound (FmVc), simple, intriguing and effective (Ho2y).

We have additionally performed experiments to address some of the evaluation concerns. Specifically, we would like to point out that our proposed method RNF does not require sensitive attribute annotation in contrast to the baseline methods like EOR and Adversarial training while performing similar or better than them in some cases. Please find below our detailed response to the questions and any concern raised by the reviewers. We will incorporate all these comments and comprehensive experimental evaluations into the revised manuscript. We are grateful to the reviewers for all the suggestions to improve our work.

Paper6943 authors

---

### Decision · Program_Chairs · 2021-09-28

**Decision:**

Accept (Poster)

**Comment:**

The paper proposes a method for debiasing classifiers. Firstly, a biased classifier is trained as normal. Then, a new classification head is trained for that encoder, that interpolates using combined representations from examples with the same label but different sensitive attributes (similarly to mix up training). This approach allows a classifier that is less biased, whilst retaining most of the performance of the original. This is clearly an important problem, and the reviewers generally found the proposed approach to be creative, soundly evaluated and well written. Therefore, I recommend acceptance.

**Consistency Experiment:**

NeurIPS has a long history of experimentation. In 2014, NeurIPS ran an experiment in which 10% of submissions were reviewed by two independent committees to quantify the randomness in the review process. This year, we repeated a variant of this experiment to see how the quality of the review process has changed over time.  This paper was part of the experiment and was therefore assigned to two committees (consisting of reviewers, an Area Chair, and a Senior Area Chair) that reached independent decisions.  If both committees made the same recommendation, this recommendation was followed. If a single committee recommended acceptance, the paper was accepted (with the exception of a few cases in which the other committee identified what we considered a fatal flaw, e.g., an error in a key result).

This copy’s committee reached the following decision: **Accept (Poster)**

The other committee assigned to the paper recommended **Reject**.  You can find the other set of reviews, along with any follow up discussion with the authors here:
https://openreview.net/forum?id=nHRGW_wETLQ